# Key Bacterial Taxa Differences Associated with Polypharmacy in Elderly Patients

**DOI:** 10.3390/microorganisms13081877

**Published:** 2025-08-12

**Authors:** Betti Shahin, Tahniat Nadeem, Tanya Khosla, Guy R. Adami

**Affiliations:** 1Department of Restorative Dentistry, College of Dentistry, University of Illinois Chicago, Chicago, IL 60607, USA; bshahi4@uic.edu; 2Department of Oral Medicine and Diagnostics, College of Dentistry, University of Illinois Chicago, Chicago, IL 60607, USA; tnade2@uic.edu (T.N.); tkhos2@uic.edu (T.K.); 3University of Illinois Cancer Center, University of Illinois Chicago, Chicago, IL 60612, USA

**Keywords:** polypharmacy, alpha diversity, aging, geriatrics, salivary microbiome, microbial dysbiosis, periodontitis, *Propionibacterium acidifaciens*, *Corynebacterium durum*

## Abstract

Changes in health, lifestyle, and medication usage significantly impact overall well-being. Aging is associated with an increased need for multiple medications, or polypharmacy. Despite extensive research on how aging and polypharmacy affect the gut microbiome, relatively little is known about their impact on the oral microbiome and how shifts there can contribute to oral and systemic disease. An initial group of 55 saliva donors was formed of individuals with stage 3 periodontal disease and well-characterized for dental decay, both factors that contribute strongly to salivary microbiome identity. Relative levels of saliva bacteria were determined by 16S rRNA amplicon sequencing. Multiple variable analysis was performed to determine taxa associated with polypharmacy after correction for dental decay, tobacco use and gender. A second group, all with stage 3 periodontal disease, over 55 years of age and controlled for caries, served as a validation set. Two differences in taxa were validated as associated with polypharmacy in the elderly group. The tooth surface commensal *Corynebacterium durum* was lower with polypharmacy, and the dental decay-associated *Propionibacterium acidifaciens* was elevated. Saliva flow rate differences did not appear to be responsible for the differences seen in these taxa. Additionally, taxa associated with caries level and gender were identified. Polypharmacy associated taxa differences are potentially directly affected by medication usage, or the ailments associated with polypharmacy, and they are strong candidates to contribute to disease in the large group of elderly with poor oral health.

## 1. Introduction

Polypharmacy is the condition of multiple medication usage that comes with pathology and also with old age. It has been defined as the usage of >4 to 6 medications [1]. It is estimated that the prevalence of polypharmacy in adults past age 65 is well over one-third [2]. It is linked to frailty, reduced survival and the presence of multiple diseases, including oral disease [3]. Polypharmacy is not only strongly negatively associated with good health, but it can also directly cause physiological changes in the body that in time can affect health in poorly understood ways [4]. Various medications have been linked to adverse events in the mouth and other sites [5,6,7]. One of the more common is lichenoid reaction—a T cell-mediated epithelial destruction associated with the usage of nonsteroidal anti-inflammatory drugs and a range of hypertensive drugs, in addition to other agents. A second common response to drugs is hyposalivation or xerostomia. It has been reported that more than half of all medications cause this effect. Given the wide range of common medications that cause lichenoid reaction and hyposalivation, polypharmacy is thought to be linked to both [5,6,8,9].

Many factors can potentially contribute in various degrees to the makeup of the oral microbiome in addition to polypharmacy. These include oral disease, particularly dental decay and periodontal disease, tobacco use, oral hygiene, and gender [10,11,12,13,14]. The presence of systemic disease can also be related to predictable differences in oral bacteria. It is difficult to differentiate the associations between systemic disease and oral microbiome from that of the medications used to treat each disease. Another layer of complication in understanding medication effects on the oral microbiome is that single medications are seldom used in isolation. Saliva represents most oral niches, so its analysis allows a window to the overall oral microbiome [10]. There is evidence from cross-sectional studies that polypharmacy itself is associated with differences in relative abundance of oral bacteria in saliva [13,15]. In some cases, these differences are linked to specific medications, but in the oral cavity, most effects are not well described or understood for the relatively common state of multiple medication usage [15].

Polypharmacy most frequently occurs with old age. With age comes increased susceptibility to periodontal disease [16]. Stage 3 periodontal disease shows high level changes in both oral bacteria and host inflammation, and it is linked to susceptibility to systemic disease [17,18,19,20,21,22,23]. The elderly also show elevated rates of hyposalivation. This study sought to determine if age and the added component of polypharmacy would be associated with additional differences in the oral microbiome. A stratified study group, all with stage 3 periodontal disease, carefully monitored for dental decay and oral soft tissue disease, was created. This allows one to minimize the direct contribution of existing oral disease to oral microbiome differences that occur with polypharmacy. Study of this group would, in theory, allow the identification of early microbiome changes associated with polypharmacy in this population at high risk for oral and systemic morbidity. People with stage 3 periodontal disease on average suffer from oral microbial imbalances, and it seemed possible that polypharmacy exacerbated that state. Alternatively, because some drugs, such as immunomodulatory drugs, are believed to lessen the effects of periodontal disease and the number of periodontal bacteria, it is also possible that polypharmacy could restore a more normal oral microbiome [24,25,26].

## 2. Materials and Methods

The study groups consisted of 55 people in the 2019 study group and 50 in the 2023 study group. They were collected consecutively as they appeared in a single general dentistry clinic one day per week at the University of Illinois Chicago between 8 November 2017 and 21 June 2019 for the 2019 group, and 4 February 2022 and 15 November 2023 for the 2023 group. Patients in the 2019 study group were drawn from a cohort of 300 patients at an inner-city dental clinic [13]. Those in the 2023 group were new samples for this study from the same clinic but collected by different clinicians. The suitability of the sample size of this group to discern a 2-fold change in relative abundance of a specific taxa between two subgroups with different levels of polypharmacy at a power of 80 and beta of 20 was calculated based on the variability in levels of the 40 most abundant taxa in the 2019 study group [27]. This analysis indicated a minimum of 36 donors total were required.

Inclusion criteria were adult, 20 or more teeth, and a recent dental and periodontal examination. Exclusion criteria were incomplete medical records, implants, acute oral disease or respiratory illness, antibiotic usage in the last month, oral antimicrobial rinse usage in the last 12 h, and food or beverage consumption in the last hour for the 2019 study group. The 2023 study group had the same inclusion criteria, with the addition of age greater than 55 and presence of stage 3 periodontal disease. Standard criteria for stage 3 severe periodontal disease were interdental clinical attachment loss of >5 mm, radiographic bone loss to the middle third of the tooth, probing depths > 6 mm, and <4 teeth lost [28]. Medication usage was gathered from the electronic health records of dental patients at the University of Illinois Chicago Dental Clinic. These were assembled by automated linkage to the patient’s pharmacy. Only medications taken long-term were included. Institutional Review Board I at the University of Illinois Chicago approved this protocol 2016-0696 and 2022-0947. All participants were informed of the implications of this study. Verbal and written consent were obtained from all participants prior to sample collection by the clinician.

### 2.1. Sample Collection, DNA Extraction, and Sequencing

Patient samples were derived from stimulated saliva accumulated from chewing paraffin wax over 5 min and processed as described earlier [29]. Saliva as collected was stored on ice for up to 2 h prior to centrifugation at 5500× *g* for 5′, followed by washing with PBS two times. DNA was extracted from frozen bacteria pellets stored at −70 °C using the Quick-DNA Fungal/Bacteria Miniprep Kit (Zymo Research, Irvine, CA, USA). The V1–V3 variable region of bacterial 16S rRNA genes was amplified using the primer set 27F/534R, followed at the University of Illinois at Chicago Sequencing Core or the Rush University Genomics and Microbiome Center by a second PCR amplification when sample-specific barcodes were added followed by cleanup and sequencing as described earlier [21]. Sequencing was performed on the Illumina MiSeq using the MiSeq reagent kit V3, 600 cycles (Illumina, Inc., San Diego, CA, USA). Negative controls were samples that started with H2O instead of saliva DNA. Additional controls were technical replicates from several donors.

For taxa assignment and measurement, forward sequence reads from the FASTQ files were analyzed using the software package QIIME2 (v2024.10) [30]. Sequences were trimmed if the average quality was lower than 25. Read sequences were initiated at 14 nt and truncated at 262 nt. DADA2-plugin in QIIME2 was used to denoise the sequence and generate feature data and feature tables for the dataset of DNA sequences [31]. Taxonomy assignment was performed by the classify-consensus-blast function using the Blast+ consensus taxonomy classifier to determine 98% match identity of the query sequences to the Human Oral Microbiome Database (v15.22) [32]. Of the 55 samples, in the 2019 group there were 38,369 reads per sample on average, ranging from 14,553 to 57,298. There were 2,148,685 reads total. In the 2024 study group, there were 50 samples plus 7 technical replicates. On average there were 32,950 reads per sample, with a range of 23,541–41,006, with 1,878,189 reads total.

### 2.2. Statistical Analysis

Statistical analysis was performed using MaAsLin2, Bioconductor version 1.22.0 software for multivariable analysis of the sample data. Compound Poisson Linear Models (CPLM) were used for the data analysis, with minimum abundance set to 2, minimum prevalence at 30%, and the FDR (*q*-value) less than 0.1 for evaluation of the 2019 study group [33]. In order to minimize false positives, the study took advantage of the model fitting failure identification of MaAsLin3, Bioconductor version 0.99.16 [34]. To do this a second model, LM for multiple variable analysis, was used. Reported are taxa that showed evidence of differential abundance using both methods, though only the CPLM output was corrected for multiple testing. For study group 2, only taxa identified as associated with medication number, caries level and gender in the first group were examined. No correction of *p*-values was needed, as only preselected taxa were queried. For statistical analysis of the study population demographics and clinical factors, descriptive statistics were used to summarize the characteristics of patients. Categorical variables were summarized using frequencies and percentages. Relationships between categorical variables were tested with the two-tailed Fisher’s exact test [35]. For discrete and continuous values, Student’s *t*-test was used to predict the significance of differences [35]. Continuous variables were presented as mean and standard deviation, and Spearman correlation coefficients were estimated to measure the relationship between two continuous or ordinal variables within R version 4.4.3 (2025-02-28).

## 3. Results

The initial study group was a subset of samples from an earlier cross-sectional study. It consisted of 55 people of various ages but all with stage 3 periodontal disease. The group included a fairly high number of tobacco users (Table 1).

The study divided this cohort into two groups based on usage of four or more medications; the polypharmacy group was significantly older (Table 1). All participants were checked for active caries. Bacterial taxa that were associated with polypharmacy were identified using MaAsLin2 multiple variable analysis. Using the CPLM analysis method, taxa were differentially abundant in those using a higher number of medications (FDR < 0.1), as shown in Table 2. This multiple variable analysis was performed while controlling for possible confounders, gender, caries level and tobacco use. The study did not attempt to control for age due to its correlation with polypharmacy [2,5]. The taxa shown in Table 2 are the 5 out of 7 that survived a second analysis with an alternate model, LM. Appendix A contain a list of the more common medications used in the populations in this study.

Taxa associated with polypharmacy, or a higher number of medications used, at an FDR of <0.1 are listed in Table 2. Taxa associated with the level of active decay sites or caries make up a long list displayed in Table 2. Finally, taxa associated with gender are also listed. An examination of the Spearman correlation between medication count and alpha diversity of the salivary microbiome revealed rs = −0.250 with *p* = 0.0697 for Shannon Diversity and rs = −0.31 with *p* = 0.0221 for Chao1 diversity for the 2019 group of participants (Figure 1).

In order to figure out how many of the taxa identified in the original dataset were truly associated with polypharmacy, a second cohort of individuals with stage 3 periodontal disease was established. It was from the same clinics, but the collection began 3 years after the earlier enrollment ended. This was an older, if not elderly, population with the additional inclusion criterion of age 55 or older. Saliva flow rates were recorded. As shown in Table 1b, only three subjects were tobacco users. Again, controlling for gender and caries level, and using multiple variable analysis, taxa associated with high medication usage were revealed. Of the five taxa that were differentially enriched with high medication usage in the first group, one was differentially abundant in the 2023 study group (Table 3). *Corynebacterium durum* was lower with polypharmacy (Figure 2). *Propionibacterium acidifaciens* trended as being higher with a probability of differential abundance of *p* = 0.097 (Figure 2). *Streptococcus mutans* just missed statistical significance as being elevated with polypharmacy in the first group and was similarly enriched in the second group with polypharmacy. A similar analysis based on gender showed the genus *Prevotella* was associated with gender in the first group, though it was not significantly differentially abundant in the second study group (Table 2). A third analysis of taxa associated with caries gave surprising results. Two taxa differentially abundant in the first group were below baseline level in the second group (meaning that they were detected in less than 30% of the participants) and were excluded. Of the remaining eight taxa, none were differentially abundant in the second group, and only *Prevotella denticola* trended as differentially abundant (Table 3).

Hyposalivation is believed to play a role in establishing oral bacterial levels and possibly relative abundances. Certain drugs, such as the anti-hypertensives, anti-depressants and more, are believed to decrease salivation rates. An examination of the relationship between saliva flow rates and medication count showed no correlation between the two in our study, likely due to the limited number of participants, 50, in the sample group. A Spearman correlation of rs = −0.031, *p* = 0.827 was observed. When multivariable analysis was used to identify the relationship between specific bacterial taxa and medication level usage, saliva flow rates showed minimal evidence for confounding. *Corynebacterium durum* was lower with effect size −0.531, standard error 0.237 at *p* < 0.0259, and *Propionibacterium acidifaciens* still trended as enriched with effect size 0.411 and standard error 0.250 at *p* < 0.113.

## 4. Discussion

Oral disease can be associated with many chronic systemic diseases, which are in turn strongly linked to medication usage; thus, one would expect polypharmacy to be associated with oral dysbiosis. However, although many drugs besides antibiotics have been shown to directly affect gut bacteria [4] and presumably cause similar changes in the mouth, there is no reason to assume the effects would be pathological. Of the two bacterial taxa validated in this study as likely being differentially abundant with multiple medication usage in elderly subjects, one, *Corynebacteria durum*, is a major structural component of plaque and is a true commensal. Its relatively low level in patients with polypharmacy may promote the formation of microcosms on teeth with more pathogenic bacteria [36,37]. The other, *Propionobacterium acidifaciens*, was enriched in the 2019 group with polypharmacy and trended as being the same in the 2023 group. It is a bacterium that is known to bind directly to dentin and contribute to dental decay [38,39,40]. Whether caused by polypharmacy, or chronic disease associated with polypharmacy, or some other characteristics of the patients, these differences can change the oral microbiome in pathogenic ways. Given that the differences occurred even after correction for oral disease, one might speculate that these are changes that precede oral disease and are firmly linked to polypharmacy.

It is already known that polypharmacy is associated with oral diseases, such as periodontal disease, caries and oral mucositis. It is thought that the first two can be caused in part by the dry mouth induced by many medications [6]. A study by DeClerq et al. identified that usage of multiple medications (two or more) results in predictable differences in abundance of certain oral genera in patients versus non-medication users. They were careful to avoid patients with serious chronic systemic disease [15]. That work suggests if multiple medication usage directly leads to oral disease, some of these taxa, including *Saprospiraceae*, which was elevated, and *Bacillus*, *Johnsonella*, *Actinobacillus*, *Stenotrophomonas* and *Mycoplasma,* which were depressed, may indeed play a role in that process. One caveat for that study is that levels of oral disease in the population were not known. Because of that, some taxa are likely differentially abundant not due to polypharmacy directly but because of the presence of oral disease in those patients. The present study has fewer participants, 105 total, but there is detailed information on oral health at the time of sample collection. In a group stratified for stage 3 periodontal disease and corrected for caries level and gender, bacterial taxa that were differentially abundant were identified. Then, that list was tested in a validation set of subjects similarly stratified for periodontal disease stage but also stratified for age and corrected for caries level. This led to the identification of *Corynebacterium durum* and *Propionibacterium acidifaciens* as possible candidates to be directly involved in polypharmacy effects. When the 2019 group was examined for genera associated with multiple medication usage in the deClerq study, only two genera were identified from that large population that were present in over 10% of the donors in our study group: *Johnsonella* and *Mycoplasma* [15]. Of these, *Johnsonella* trended as showing a lower level in the multi-medication group, *Johnsonella* at coefficient −0.0967 and *p* < 0.063, in agreement with the earlier study. *Mycoplasma* did not. It should be noted that it is equally possible that the differential levels of these taxa in those with high levels of medications are related to the need for polypharmacy or the presence of chronic diseases for which the drugs are prescribed.

Studies on frailty may be relevant to our work. It is to be noted that there was a modest but significant negative correlation between Chao1 alpha diversity (richness) and medication level for the 2019 group, R(53) = −0.25, *p* < 0.026, though it did not reach statistical significance for the 2023 group. Given Langille’s observation that fragility correlated with reduced alpha diversity, this is in general agreement with that result, as polypharmacy is known to correlate with fragility [41]. Many medications have been linked to hyposalivation, and both are associated with loss of fitness. However, contrary to what one might expect, there was little loss of significance of association between the two taxa and polypharmacy when correcting for saliva flow rates. This would suggest stimulated saliva flow rate is not a major factor establishing these associations in this population.

There was little overlap between the caries-associated taxa in the 2019 and 2024 study groups. Many of the salivary taxa identified as caries-associated in the 2019 group were not even present at baseline levels in the 2023 group. This may in part be due to higher levels of decay in the 2019 study group. An examination of caries associated with the 2019 group is like a who’s who of cariogenic/caries-associated bacteria [42]. Out of the 10 total taxa, 3 were of the *Prevotella* genus, 2 of *Actinomyces*, and the species *Streptococcus mutans* was identified as caries-associated. None of these were differentially abundant with caries in the 2023 group. One, though, *Prevotella denticola*, trended as being at higher levels with caries. Another difference was that the 2023 group was significantly older, >55 years, than the 2019 group. Interestingly, the bacteria that were preliminarily identified as enriched with high caries levels in the 2023 group have been found to be enriched in populations with root caries—an ailment chiefly of the elderly [43]. This included *Leptotrichia* and *Fusobacterium nucleatum subsp._animalis.*

Limitations of this study are the relatively small N and the usage of a single facility to collect all samples. However, the fact that the two patient groups were separated by over 3 years, the COVID-19 pandemic and used different clinicians and donors increases the independence of the datasets. It was not expected that the second group of patients’ oral sample microbiome profiles would show higher levels of variability. This may be a product of the variability in oral microbiome seen in elderly patients that others have noted [8]. This did not preclude us from using the validation set to verify as differentially abundant the short list of taxa identified by analysis of the 2019 dataset. A future goal is to provide more detail on this phenomenon. This would include, if applicable, identifying medications that contribute directly to alterations of oral bacteria.

## 5. Conclusions

This study does not suggest that the results here differentiate bacteria taxa associated with polypharmacy from those of multimorbidity. All analyses were corrected for oral disease, which may prevent us from detecting some bacterial taxa that are associated with polypharmacy and directly and rapidly cause oral disease. However, given the slow onset of oral disease under most circumstances, it increases the possibility that the bacterial taxa shown to be elevated or decreased with polypharmacy could in time lead to oral and possibly worsening systemic disease. Lower levels of *Corynebacteria durum* and higher levels of *Propionibacterium acidifaciens* would theoretically contribute to oral pathology in these people.

## Figures and Tables

**Figure 1 microorganisms-13-01877-f001:**
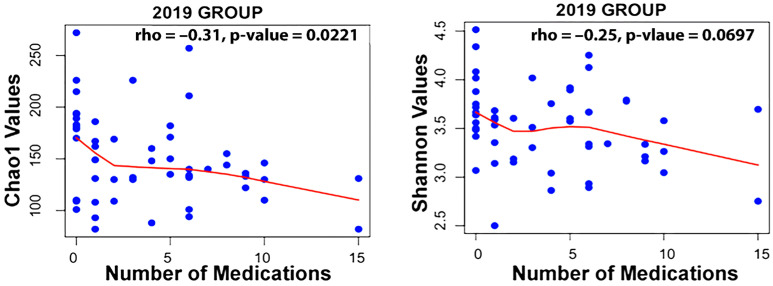
Correlation between medication count and alpha diversity for the 2019 dataset using the Spearman method.

**Figure 2 microorganisms-13-01877-f002:**
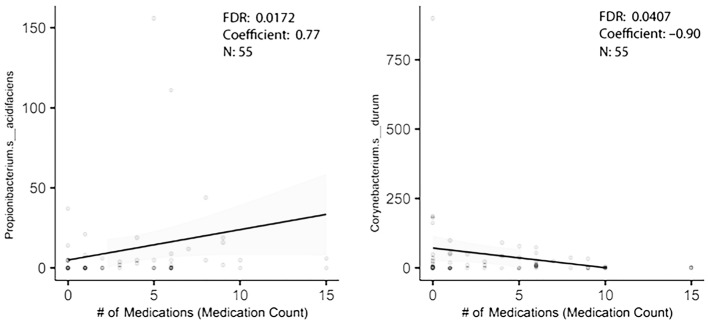
MaAsLin2 analysis reveals two taxa differentially abundant with polypharmacy.

**Table 1 microorganisms-13-01877-t001:** Demographics and clinical characteristics of study groups. (a) Demographic and clinical characteristics of the 2019 group. (b) Demographic and clinical characteristics of the 2023 group.

(a)			
Characteristics (2019 Group)	Polypharmacy ≤ 3 Drugs *n* = 29	Polypharmacy ≥ 4 Drugs *n* = 26	*p*-Value ^1^ (1.0)
Age, years (mean ± SD)	51.5 ± 14.8	62.7 ± 8.64	1.60 × 10^−3^
Sex, male *n* (%)	11.0 (37.9)	11.0 (42.3)	4.00 × 10^−1^
Caries ^2^, *n* (mean ± SD)	4.07 ± 6.87	6.35 ± 7.83	2.60 × 10^−1^
Medications, *n* (mean ± SD)	0.793 ± 1.01	7.28 ± 3.01	1.00 × 10^−4^
Tobacco User, *n* (%)	6.00 (20.7)	2.00 (0.077)	3.30 × 10^−1^
**(b)**			
**Characteristics (2023 Group)**	**Polypharmacy ≤ 3 Drugs** ***n* = 26**	**Polypharmacy ≥ 4 Drugs** ***n* = 24**	***p*-Value ^1^** **(1.0)**
Age, years (mean ± SD)	68.1 ± 7.44	72.2 ± 8.33	7.70 × 10^−2^
Sex, male *n* (%)	9.00 (34.6)	11.0 (45.8)	6.00 × 10^−1^
Caries ^2^, *n* (mean ± SD)	2.00 ± 2.10	1.79 ± 2.70	7.60 × 10^−1^
Medications, *n* (mean ± SD)	1.42 ± 1.24	5.96 ± 2.31	1.00 × 10^−4^
Saliva ^3^, mL (mean ± SD)	3.91 ± 1.51	2.70 ± 3.25	7.00 × 10^−1^
Tobacco User, *n* (%)	1.00 (0.038)	1.00 (0.042)	1.00

^1^ *p*-value for age, caries, and medications was determined using Student’s *t*-test; *p*-values for sex and tobacco use were determined using Fisher’s exact test. ^2^ Caries was measured as the number of decayed tooth surfaces. ^3^ Saliva was stimulated, and the volume was collected in 3 min.

**Table 2 microorganisms-13-01877-t002:** Association of various factors with saliva microbiome for the 2019 group in the multivariable association model (MaAsLin2 analysis).

Taxa associated with Medication Count ^1^					
Taxa	Coefficient ^2^	Standard Error ^3^	Samples not 0	*p*-value	*q*-value
*Propionibacterium.s_ acidifaciens*	0.773	0.205	27	1.59 × 10^−4^	1.72 × 10^−2^
*Leptotrichia.s_sp._HMT_212*	−0.779	0.209	36	1.88 × 10^−4^	1.73 × 10^−2^
*Corynebacterium.s_ durum*	−0.901	0.273	35	9.70 × 10^−4^	4.08 × 10^−2^
*Capnocytophaga.s_gingivalis*	−0.736	0.233	37	1.61 × 10^−3^	5.15 × 10^−2^
*Atopobium.s_rimae*	0.518	0.168	40	2.09 × 10^−3^	5.71 × 10^−2^
**Taxa associated with Gender ^4^**					
Taxa	Coefficient ^2^	Standard Error ^3^	Samples not 0	*p*-value	*q*-value
*g_Prevotella._*	−0.788	0.232	54	6.91 × 10^−4^	3.79 × 10^−2^
**Taxa Associated with Caries Level ^5^**					
Taxa	Coefficient ^2^	Standard Error ^3^	Samples not 0	*p*-value	*q*-value
*Propionibacteium.s_acidifaciens*	0.809	0.182	27	9.02 × 10^−6^	3.32 × 10^−3^
*Actinomyces.s_dentalis*	0.558	0.124	34	6.80 × 10^−6^	3.32 × 10^−3^
*Prevotella.s_sp._HMT_292*	0.600	0.141	30	2.19 × 10^−5^	5.37 × 10^−3^
*Prevotella.s baroniae*	0.644	0.168	24	1.21 × 10^−4^	1.72 × 10^−2^
*Streptococcus.s _mutans*	0.822	0.243	38	7.21 × 10^−4^	3.79 × 10^−2^
*Streptococcus.s_anginosus*	0.610	0.178	38	6.26 × 10^−4^	3.79 × 10^−2^
*Prevotella.s_denticola*	0.515	0.159	45	1.16 × 10^−3^	4.07 × 10^−2^
*Saccharibacteria_.TM7._.G.6..s_bacterium_HMT_870*	−1.01	0.312	34	1.14 × 10^−3^	4.07 × 10^−2^
*Gemella._*	0.633	0.218	40	3.73 × 10^−3^	8.85 × 10^−2^
*Actinomyces. s sp. HMT 448*	0.566	0.196	37	3.86 × 10^−3^	9.50 × 10^−2^

^1^ Multivariable analysis adjusted for gender, caries level, and tobacco use. ^2^ Coefficient refers to the model coefficient or effect size. ^3^ Standard error is for the standard error of the mean. ^4^ Multivariable analysis adjusted for caries level, medication count, and tobacco use. ^5^ Multivariable analysis adjusted for gender, medication count, and tobacco use.

**Table 3 microorganisms-13-01877-t003:** External validation of taxa associations using the 2023 group.

Validation of Taxa Associated with Medication Count ^1^				
Taxa	Coefficient	Standard Error	Samples not 0	*p*-value
*Propionibacterium.s_ acidifaciens*	0.422	0.255	25	9.72 × 10^−2^
*Leptotrichia.s_sp._HMT_212*	−0.193	0.233	40	4.07 × 10^−1^
*Corynebacterium.s_ durum*	−0.510	0.241	38	3.48 × 10^−2^
*Capnocytophaga.s_ gingivalis*	−0.299	0.240	43	2.14 × 10^−1^
*Atopobium.s_rimae*	0.121	0.259	29	6.39 × 10^−1^
**Validation of Taxa Associated with Gender ^2^**				
Taxa	Coefficient	Standard Error	Samples not 0	*p*-value
*g_Prevotella._*	−0.0933	0.277	55	2.77 × 10^−1^
**Validation of Taxa Associated with Caries Level ^3^**				
Taxa	Coefficient	Standard Error	Samples not 0	*p*-value
*Propionibacteium.s_acidifaciens*	0.271	0.249	25	2.77 × 10^−1^
*Actinomyces.s_dentalis*	0.00438	0.269	33	9.87 × 10^−1^
*Prevotella.s_sp._HMT_292*	ND	ND	ND	ND
*Prevotella.s baroniae*	ND	ND	ND	ND
*Streptococcus.s _mutans*	0.161	0.305	34	5.97 × 10^−1^
*Streptococcus.s_anginosus*	0.229	0.321	37	4.76 × 10^−1^
*Prevotella.s_denticola*	0.390	0.236	33	9.84 × 10^−2^
*Saccharibacteria_.TM7._.G.6..s_bacterium_HMT_870*	−0.396	0.379	27	2.96 × 10^−1^
*Gemella._*	0.0595	0.188	54	7.52 × 10^−1^
*Actinomyces. s sp. HMT 448*	0.0634	0.435	31	8.84 × 10^−1^

^1^ Multivariable analysis adjusted for gender and caries level. ^2^ Multivariable analysis adjusted for caries level and medication count. ^3^ Multivariable analysis adjusted for gender and medication count.

## Data Availability

The original contributions presented in this study are included in the article/Appendix A. Further inquiries can be directed to the corresponding author.

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
