# Peer review of "Key Bacterial Taxa Differences Associated with Polypharmacy in Elderly Patients"

_microorganisms, 2025, doi:10.3390/microorganisms13081877_

Round 1
Reviewer 1 Report
Comments and Suggestions for Authors
see enclosed pdf

Reviewer 2 Report
Comments and Suggestions for Authors
I thank authors for conducting such studies. The only concerns I have was the mixed data included. Yet, statistical analysis tried to lessen the variabilities inherited. I suggest that the title may bay be modified to reflect the overall outcome like "Impact of Polypharmacy on Key Bacterial Taxa in elderly patients". Also, change the word "decay" to caries in manuscript.
Round 2
Reviewer 1 Report
Comments and Suggestions for Authors
The manuscript has been improved